# Damage Analysis via Bidirectional Multi-Task Cascaded Multimodal Fusion

## Abstract

Damage analysis in social media platforms such as Twitter is a comprehensive problem which involves different subtasks for mining damage-related information from tweets (*e.g.*, informativeness, humanitarian categories and severity assessment). The comprehensive information obtained by damage analysis enables to identify breaking events around the world in real-time and hence provides aids in emergency responses. Recently, with the rapid development of web technologies, multimodal damage analysis has received increasing attentions due to users' preference of posting multimodal information in social media. Multimodal damage analysis leverages the associated image modality to improve the identification of damage-related information in social media. However, existing works on multimodal damage analysis address each damage-related subtask individually and do not consider their joint training mechanism. In this work, we propose the Bidirectional Multi-task Cascaded multimodal Fusion (BiMCF) approach towards joint multimodal damage analysis. To this end, we introduce the cascaded multimodal fusion framework to separately integrate effective visual and text information for each task, considering that different tasks attend to different information. To exploit the interactions across tasks, bidirectional propagation of the attended image-text interactive information is implemented between tasks, which can lead to enhanced multimodal fusion. Comprehensive experiments are conducted to validate the effectiveness of the proposed approach.

## CCS Concepts

• **Human-centered computing → Social media**; • **Information systems → Multimedia information systems**.

## Keywords

Damage analysis, Social network Analysis, Feature fusion, Multimodal deep learning, Multi-task learning

**ACM Reference Format:**
Anonymous Author(s). 2024. Damage Analysis via Bidirectional Multi-Task Cascaded Multimodal Fusion. In Proceedings of The Web Conference 2025. ACM, New York, NY, USA, 10 pages. https://doi.org/10.1145/nnnn

## 1 Introduction

Over the past years, social media platforms such as Twitter have become more and more popular in people's daily life. With the

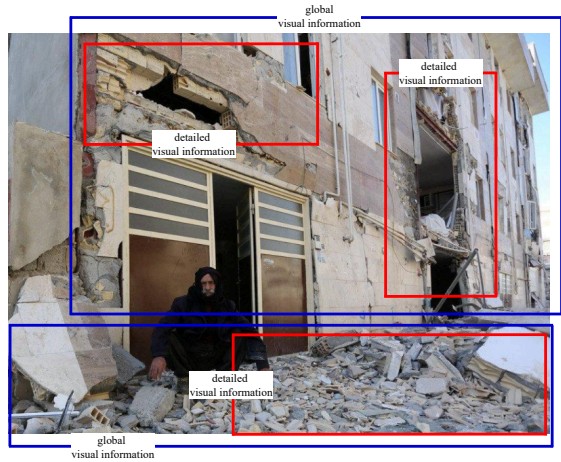

Putin expressed his condolences to President of Iran because of the earthquake: https://t.co/4hwqdJtoBD https://t.co/iayi92AXv6 #Russia

**Figure 1: A damage analysis sample. The attended visual regions by different subtasks are marked with different colors. The blue box represents the informativeness subtask, while the red box represents the event classification subtask.**

rapid development of web technologies, people tend to share their daily-life and news on social media in real-time. Recently, social medias have been exploited to identify breaking events around the world [1, 3, 8, 9, 34, 36, 48], due to their increasing popularity. Event detection in social media can provide real-time emergency response to emergency events (*e.g.*, natural disasters, vehicle damages and missing people) happened all around the world. In general, emergency event detection involves comprehensive damage analysis consisting of several subproblems, *e.g.*, informativeness prediction, event classification and severity assessment [1].

The early works on social media damage analysis mainly focus on mining damage-related information solely from texts [5, 40]. Recently, multimodal damage analysis has received increasing attentions due to people's preference of posting multimodal information in social media [1, 4, 19]. Therefore, the CrisisMMD dataset is created as standard benchmark for multimodal damage analysis [4]. Samples of CrisisMMD are collected by crawling blogs posted on Twitter in times of seven natural disasters (*e.g.*, floods, wildfires, hurricanes, earthquakes) and featured by three class labels (*i.e.*, informativeness, humanitarian categories and severity). These class labels follow a cascaded structure, where an posted blog must be identified as informative before it can be classified into a specific humanitarian category and only after this classification can the severity of the event be assessed. Afterwards, various multimodal learning methods are proposed for multimodal damage analysis based on the CrisisMMD benchmark [1, 2, 19, 21, 31]. Compared

with unimodal methods, multimodal damage analysis leverages the associated image modality to improve the identification of damage-related information in social media [20].

However, most of existing works on multimodal damage analysis address each damage-related subtask independently and do not consider to leverage the inherent relationships across tasks [8, 26, 28, 35]. From the perspective of multi-task learning [24, 27, 50], a joint training framework can lead to improved performance for each task by exploiting the relationships between tasks. Although a multi-task damage assessment system has been designed in [2], it only organizes subtasks in a pipelined way and does not consider the joint training of them. The motivation of jointly training damage analysis subtasks establishes the multimodal multi-task learning framework. Similar to general multi-task learning methods [50, 52], existing multimodal multi-task methods mainly focus on learning shared multimodal representations [13, 32, 49], which effectively facilitate information sharing between different tasks. However, they only implicitly reflect task correlations, making it less effective when there are clear relationships among tasks. Figure 1 displays a multimodal damage analysis sample about two cascaded tasks and we can observe that informativeness prediction and event classification subtasks attend to different image-text interactive information. Specifically, the informativeness prediction task pays attention to global features, while the event classification task further narrows down to specific detail features based on the attended information by the informativeness prediction task. The shared multimodal representations can't explicitly explore such intrinsic interdependencies and it is also challenging to meet the feature requirements of all tasks, thus leading to degraded performance.

In this work, we propose the Bidirectional Multi-Task Cascaded multimodal Fusion (BiMCF) approach towards joint multimodal damage analysis. Different from previous works which implement multi-task learning based on shared multimodal representations, our approach deploys the multi-task interaction mechanism in the implementation of multimodal fusion. Following [1, 15, 46], we adopt BERT as the main backbone and integrate visual information into token representations based on cross-modal attention introduced in [1]. For different tasks, we introduce the cascaded multimodal fusion framework to separately integrate visual and text information for each task, considering the fact that different tasks attend to different information. Furthermore, the forward and backward gate mechanism is introduced in our bidirectional propagation framework for each task to enhance task-specific feature extraction and facilitate effective multimodal fusion. Specifically, the forward gate concentrates on amplifying salient image features to provide more detailed information, while the backward gate focuses on expanding attention regions to offer richer global information. Then the multimodal transformer is applied to update text features. Therefore, our bidirectional cascaded structure can alleviate errors accumulated in one direction and enhance the identification of damage-related information. Unlike the prior works [6, 25, 43] which consider multi-task cascades in the final prediction of labels, our approach deploys the cascaded structure of tasks into the multimodal fusion process of each task, thereby explicitly encoding the interdependencies among subtasks into the model structure. To this end, the cross-modal attention incorporates information from both textual and visual modalities more effectively.

To sum up, the contributions of this work are mainly three-fold:

- We propose to implement comprehensive multimodal damage analysis from the multi-task learning perspective, with the purpose of exploiting the potential relationships between tasks.
- We propose the BiMCF approach by introducing the bidirectional cascaded framework to implement multimodal multi-task learning, which encodes the inderpendencies among subtasks into the model structure and enhance the multimodal fusion process.
- Comprehensive experiments on the CrisisMMD benchmark are conducted to show the effectiveness of our approach. Our approach can achieve better performance than existing works.

## 2 Related Work

### 2.1 Damage Analysis

Damage analysis in social media can provide real-time emergency response to emergency events. The previous works on damage analysis mainly focus on mining damage-related information solely from images [18, 30] or texts [5, 40, 41]. Recently, attention has been paid to multimodal damage analysis due to people's increasing interests in posting multimodal messages on social media platforms. In particular, Alam et al. [4] create the CrisisMMD dataset as standard benchmark for multimodal damage analysis by crawling blogs from Twitter. Agarwal et al. [2] propose the Crisis-DIAS framework towards comprehensive multimodal damage analysis based on CrisisMMD. Recently, Liang et al. [21] consider fine-grained cross-modal interactions in feature fusion and obtain improved damage categorization performance. Mariham et al. [38] integrate the adaptive attention mechanism with late fusion strategy to achieve effective cross-modal interaction. Mohammad et al. [8] propose a sequential hierarchical framework to classify social media information, and employ different models to effectively process multimodal data. Furthermore, Bishwas et al. [26] propose to fine-tune pre-trained contrastive models to distinguish multimodal crisis tweets. However, existing works address each damage-related subtask individually and do not exploit the cascaded task relationships for training from a multi-task learning perspective.

### 2.2 Multimodal Multi-Task Learning

Multi-task learning exploits task relationships to regularize the training of deep models. Most of existing works implement multi-task learning based on task-shared representations. In multimodal settings, existing works also build on task-shared multimodal representations [25, 29, 43, 49], where features of different modalities are firstly fused into a multimodal representation which is then utilized to implement multi-task learning. Following this framework, Tan et al. [43] propose the MultiCoFusion model for cancer prognosis prediction from multi-modal inputs. Their approach performs alternate training between different tasks on task-shared multimodal representations. Alam [6] proposes the Multimodal Spatiotemporal Neural Fusion network for Multi-Task Learning, in which multi-task cascaded learning are developed on the fusion layer. Maity et al. [25] implement both FeedBack Multi-task learning and Central-Net Multi-task learning on fused features. Zhang et al. [49] propose to share multimodal features on a sparse space, ensuring that only features beneficial to decision-making are shared. In general, existing works do not consider the uniqueness and correlation of

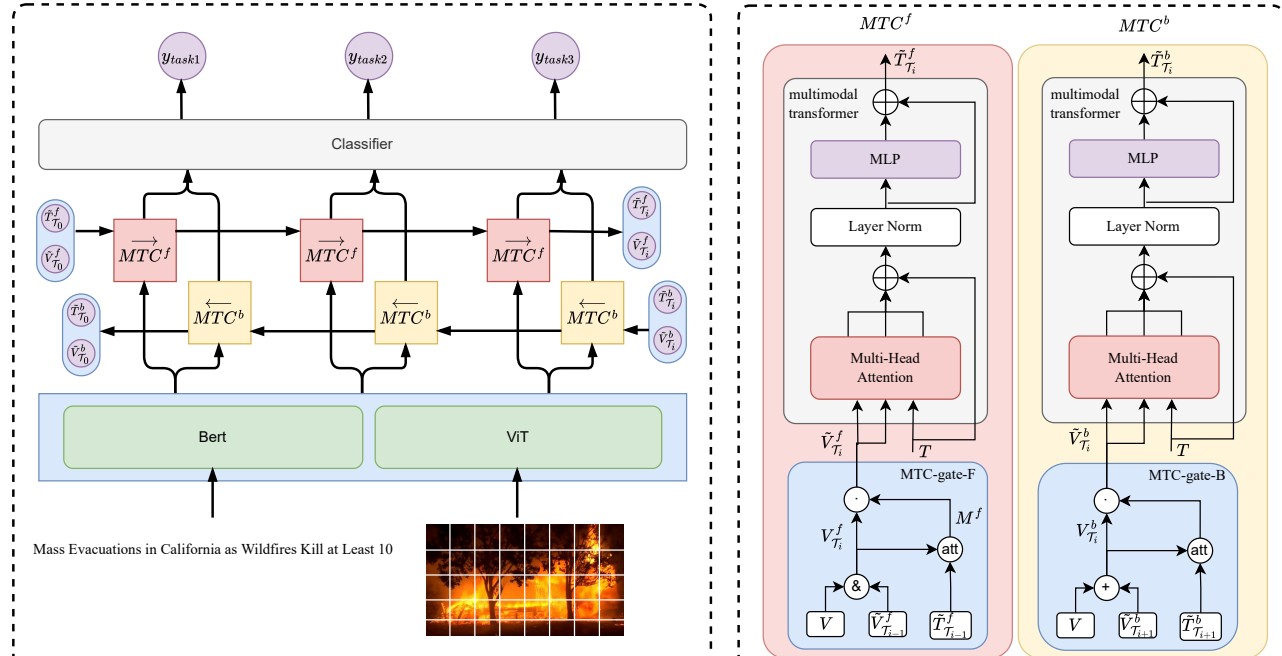

**Figure 2: (Left) The overall architecture of the proposed BiMCF approach. First, for a pair of text and image input, we use BERT and ViT to generate feature representations respectively. Then the image and text features are passed and updated through $MTC^f$ and $MTC^b$. Last, the final prediction for each task $\mathcal{T}_i$ is decided by two output text features from both directions, namely $T^b_{\mathcal{T}_i}$ and $T^f_{\mathcal{T}_i}$. (Right) The detailed frameworks of $MTC^f$ and $MTC^b$. '&' demotes max operator and '+' denotes addition operator.**

different tasks during multimoddal fusion. Different from existing works which only leverage task relationships for final predictions, this work further considers to exploit multi-task interactions in the implementation of multimodal fusion and introduces a more reasonable multi-task cascaded multimodal fusion framework.

## 3 METHODOLOGY

### 3.1 Problem Definition

In general, damage analysis involves three related subtasks [1, 2, 4, 21], namely informativeness prediction, event classification and severity prediction. We adopt the same label settings used in [21]. In particular, the informativeness prediction task identifies whether social media posts can provide useful damage-related information for emergency response. For informative social media posts, the event classification task further identifies the damage types. Finally, the severity prediction task assesses the severity level of damages reported in social media posts. The above three tasks follow the cascaded relationship, where only an informative post can be classified into a specific humanitarian category and only after this classification can the severity of the event be assessed.

In a multimodal setting, we are given a pair of text and image input $X = \{T, V\}$ where $T$ denotes a text sequence, and $V$ denotes an image. The outputs consist of three labels $Y = \{y_I, y_H, y_S\}$ where $y_I$ denotes whether the input is informative, $y_H$ refers to the humanitarian category of $X$, and $y_S$ indicates the severity level of the event. We denote the training data consisting of both paired text and image input as well as a label set for all the three subtasks

as $\mathcal{D} = \{(X_i, Y_i)\}_{i=1}^N$. During inference, the target is to predict the set of labels $Y_j$ for each test instance $X_j$.

### 3.2 Motivation & Model Framework

Multi-task frameworks have been widely used to solve problems involving multiple tasks in a joint manner. These frameworks include pipelined approaches [17, 44] in which some tasks are performed in the first place and their outputs are passed to the remaining tasks. This operation results in error propagation when the model does not perform well on earlier tasks. Another strategy adopts (hard or soft) parameter sharing [27, 39] among different tasks with the intuition that the shared parameters encode common features or relationships across tasks. However, parameter sharing only implicitly reflects task correlations, making it less effective when there are clear relationships among subtasks, which is the case for multi-level damage analysis. Specifically, the three subtasks form a cascade structure in which the class label of each task greatly relies on the prediction of its preceding task.

To explicitly infuse the evolution of the involved tasks, we propose a bidirectional multi-task cascaded framework which ranks the three subtasks in the order of informativeness prediction, humanitarian categorization and severity level prediction based on the granularity (coarse to fine) of the problem. In the three cascaded tasks, the later task further narrows down to specific detailed features based on the attended information by its preceding task. Therefore, the early tasks are more coarse-grained while the later tasks are more fine-grained. This sequential approach helps the model gradually understand detailed information. Additionally, we

have validated the superiority of our task sequence in section 4. Inspired from the working mechanism of a zoom camera which narrows the range of the lens but amplifies the focused area each time it zooms in, we treat the ordered task cascade as a zooming process. In a forward pass, we sequentially zoom in images based on the output from a preceding task. On the other hand, an addition of a backward pass performs the opposite operation which zooms out an image given the output from a finer-grained task. This bidirectional cascade structure propagates information in dual directions to alleviate errors accumulated in one direction. Meanwhile, the bidirectional flows effectively exchange information from related tasks to enhance a better prediction for the target subtask.

To incorporate information from both textual and visual modalities, we further adopt a unified multimodal fusion framework across all the tasks. Specifically, we treat text representations as a common reference which is used to guide the attention computations over the resulting image representations at each zooming step. The final multimodal features for each task are obtained via cross-modal transformers. Using text modality as a reference is attributed to the fact that information is usually more explicitly expressed in natural languages. To this end, we propose a bidirectional multi-task cascaded framework with multimodal fusion to solve the three subtasks of damage analysis in an end-to-end manner.

The overall architecture of BiMCF is shown in Figure 2. Specifically, given an input textual description $T$ and an image $V$, we first generate a feature representation for each token in the text as $\mathbf{T} = [\mathbf{t}_1, \mathbf{t}_2, ..., \mathbf{t}_n]$ using a pretrained language model, $e.g.$, BERT [10]. The image feature representations are produced following the existing works using ViT [11]. In particular, the image is first resized to 224×224 and then divided into $m$ image patches which are further reshaped into a sequence. We extract the features from the last layer of ViT as visual input representations, denoted as $\mathbf{V} = [\mathbf{v}_1, .\mathbf{v}_2, ..., \mathbf{v}_m]$. Here we define $\mathbf{T} \in \mathbb{R}^{n \times D_t}$ and $\mathbf{V} \in \mathbb{R}^{m \times D_v}$. BiMCF processes the input features $\mathbf{T}$ and $\mathbf{V}$ in two opposite directions, both consisting of three modules corresponding to three subtasks. Each $i$-th module incorporates a Multimodal Task Cascading (MTC) gate which takes the original image features $\mathbf{V}$, the output image features $\tilde{\mathbf{V}}_{\mathcal{T}_{i-1}}$ and the output text features $\tilde{\mathbf{T}}_{\mathcal{T}_{i-1}}$ from its preceding task as input to decide the information flow from its preceding task. We design different gate operations for forward pass (denoted as $\mathcal{G}_{MTC-f}$) and backward pass (denoted as $\mathcal{G}_{MTC-b}$), simulating the zoom-in and zoom-out mechanism, respectively. The output $\tilde{\mathbf{V}}_{\mathcal{T}_i}$ from each MTC gate is then treated as the updated image representation to be fused with the text representation $\mathbf{T}$ via a multimodal transformer block. We use the final output text features $\tilde{\mathbf{T}}_{\mathcal{T}_i}$ after the multimodal fusion process for task predictions. The details of each component will be further illustrated in the following sections.

### 3.3 Task-Specific Multimodal Transformers

For each task, we exploit the complex interactions between the textual modality and the visual modality using multimodal transformers. Specifically, for each task $\mathcal{T}_i$, given input features for a sentence $\mathbf{T} = [\mathbf{x}_1, \mathbf{x}_2, ..., \mathbf{x}_n]$ and an image $\mathbf{V} = [\mathbf{v}_1, .\mathbf{v}_2, ..., \mathbf{v}_m]$, the multimodal transformers construct its query matrix using the text features $\mathbf{TW}_q$. The key and value matrices are generated based on image features $\mathbf{VW}_k$ and $\mathbf{VW}_v$, respectively, where $\mathbf{W}_q \in$

$\mathbb{R}^{D_t \times D}$, $\mathbf{W}_k, \mathbf{W}_v \in \mathbb{R}^{D_v \times D}$ are transformation matrices. The cross-modal attentions then compute the attention weights over image patches and generate an updated output matrix as:

$$\mathbf{H} = \text{softmax}\left(\frac{(\mathbf{TW}_q)(\mathbf{VW}_k)^\top}{\sqrt{D}}\right)(\mathbf{VW}_v). \tag{1}$$

This completes a single-head cross-modal attention computation. With multiple heads, the final output text representation, denoted as $\tilde{\mathbf{T}}$, is computed as

$$\tilde{\mathbf{T}} = f_{LN}(f_{FF}([\mathbf{H}_1; ...; \mathbf{H}_M]) + \mathbf{T}). \tag{2}$$

Here $f_{LN}$ and $f_{FF}$ denote layer norm and feed-forward functions, respectively. $M$ denotes the number of attention head. We use different transformer parameters for different subtasks and denote by $\tilde{\mathbf{T}}_{\mathcal{T}_i} = \mathcal{M}_i(\mathbf{V}, \mathbf{T})$ as the multimodal transformer function for task $\mathcal{T}_i$ obtained through equation (1) and (2).

### 3.4 Multimodal Task Cascade

Task-specific multimodal transformers only exploit cross-modal interactions, but treat each task independently, ignoring the essential relationships among different tasks. In this aspect, multitask frameworks are commonly used to exploit task relationships for learning and allocating shared representations. However, it is not clear whether these sharing strategies align with the actual task relationships and how they promote task correlations. In this work, we target on multi-task setting with a cascaded structure and explicitly encode such relation into the model architecture via Multimodal Task Cascading (MTC) gate. To capture this dependency and ordered relationship between tasks, we use MTC gates within each task module to explicitly update the features of the input image. Different from common practices that use the same image input for multiple tasks, in our work, we feed different tasks with distinct image features generated from the MTC gate. In what follows, we introduce two gating mechanisms adopted in forward information flow and backward information flow, respectively.

**Forward Gate.** In the forward flow, we order the three tasks according to the increasing granularity level, namely informativeness prediction, humanitarian categorization, followed by severity level prediction. The later task requires more fine-grained analysis of the input than its preceding task and could benefit from the features of its preceding task. For example, suppose task $\mathcal{T}_i$ aims to predict whether there is an event and is provided with a scaled image focusing on the upper right corner. When the subsequent task $\mathcal{T}_{i+1}$ aims to predict the exact event type, it could be beneficial to zoom in the upper right corner of the image to look for more salient evidence, because the rest parts of the image are not relevant to events. Hence, the objective of the forward gate is to further amplify the salient features in each image based on the selected regions generated in the preceding task. This is similar to a zoom-in process of a camera which first locates a more global area of an image and amplifies this specific area for a deeper examination.

Formally, for task $\mathcal{T}_i$, the forward gate considers the interaction of the current task and its preceding task $\mathcal{T}_{i-1}$. Given the original image features $\mathbf{V}$, the output text features $\tilde{\mathbf{T}}^f_{\mathcal{T}_{i-1}}$ and image features $\tilde{\mathbf{V}}^f_{\mathcal{T}_{i-1}}$ computed from the module of task $\mathcal{T}_{i-1}$ in the forward pass, the forward MTC gate computes a weight score $s^f_j$ for each image

patch to scale its effect prior to the multimodal fusion process. Here we use the superscript $f$ to denote the forward pass. Specifically, we first apply a max operator over two image features $\mathbf{V}$ and $\tilde{\mathbf{V}}^f_{\mathcal{T}_{i-1}}$, resulting in $\mathbf{V}^f_{\mathcal{T}_i} = \max(\mathbf{V}, \tilde{\mathbf{V}}^f_{\mathcal{T}_{i-1}})$, where max is an element-wise selector. This operation mimics the zoom-in effect which tries to keep salient features obtained in the preceding task. On the other hand, the preceding task may propagate errors that negatively affect the current task. The max operator could mitigate this issue by retaining salient features from the original image.

The second step computes a scaling score for each image patch according to:

$$\mathbf{M}^f = \mathcal{L}^f_1(\mathbf{V}^f_{\mathcal{T}_i}) \cdot \mathcal{L}^f_2(\tilde{\mathbf{T}}^f_{\mathcal{T}_{i-1}})^\top, \tag{3}$$

$$s^f_j = \frac{\exp^{\sum_{k=1}^n \mathbf{M}^f_{[j,k]}}}{\sum_{j'=1}^m \exp^{\sum_{k=1}^n \mathbf{M}^f_{[j',k]}}}, \tag{4}$$

where $\mathcal{L}^f_1$ and $\mathcal{L}^f_2$ denote two linear transformation functions used to project the inputs $\mathbf{V}^f_{\mathcal{T}_i}$ and $\tilde{\mathbf{T}}^f_{\mathcal{T}_{i-1}}$ into $\mathbb{R}^{m \times D_v}$ and $\mathbb{R}^{n \times D_v}$, respectively. Each element $\mathbf{M}^f_{[j,k]}$ in $\mathbf{M}^f \in \mathbb{R}^{m \times n}$ stores the correlation score between the $j$-th image patch and the $k$-th text token. This reflects cross-modality correlations and treats the text modality as a reference to guide the zooming process. Equation (4) applies a softmax operation over the sum of correlation scores of each image patch with text tokens. A larger score implies higher correlations of its corresponding image patch with the text input, and thus implies higher chance of being retained for the target task. The final output image features are

$$\tilde{\mathbf{V}}^f_{\mathcal{T}_i} = s^f \odot \mathbf{V}^f_{\mathcal{T}_i}, \tag{5}$$

where $\odot$ denotes row-wise multiplications. An exception happens to the first subtask $\mathcal{T}_1$ where there is no preceding task for guidance. In this case, we use $\tilde{\mathbf{T}}^f_{\mathcal{T}_0} = \mathbf{T}$ and $\tilde{\mathbf{V}}^f_{\mathcal{T}_0} = \mathbf{V}$ as the input to the forward gate. For ease of notation, we use $\tilde{\mathbf{V}}^f_{\mathcal{T}_i} = \mathcal{G}_{\text{MTC}-f}(\mathbf{V}, \tilde{\mathbf{T}}^f_{\mathcal{T}_{i-1}}, \tilde{\mathbf{V}}^f_{\mathcal{T}_{i-1}})$ to denote all the above-mentioned computations for the forward gate.

**Backward Gate.** From an opposite perspective as the forward gate, we additionally consider the backward information flow which starts from the most fine-grained subtask, *i.e.*, severity level prediction, followed by humanitarian categorization and informativeness prediction. Generally the finer-grained task provides a small range of focused area in an image that is most effective for predictions. It is then beneficial to start from this precise area and gradually expand the region around it for subsequent tasks which are more coarse-grained. This assembles the zoom-out process of a camera which helps to expand the search space. Hence, the objective of the backward gate is to pivot image attentions around some anchor regions obtained from the preceding task.

Formally, for task $\mathcal{T}_i$, the backward gate $\mathcal{G}_{\text{MTC}-b}$ considers the interaction of $\mathcal{T}_i$ and $\mathcal{T}_{i+1}$. The input of the backward gate consists of the original image features $\mathbf{V}$, the output text features $\tilde{\mathbf{T}}^b_{\mathcal{T}_{i+1}}$ and image features $\tilde{\mathbf{V}}^b_{\mathcal{T}_{i+1}}$ from task $\mathcal{T}_{i+1}$, in which the superscript $b$ denotes the backward pass. Different from the forward gate, the first step of the zoom-out process uses element-wise addition to simulate the expanding effect: $\mathbf{V}^b_{\mathcal{T}_i} = \mathbf{V} + \tilde{\mathbf{V}}^b_{\mathcal{T}_{i+1}}$. Similar to the forward

gate, the second step then computes the scaling score $s^b$ as follows:

$$\mathbf{M}^b = \mathcal{L}^b_1(\mathbf{V}^b_{\mathcal{T}_i}) \cdot \mathcal{L}^b_2(\tilde{\mathbf{T}}^b_{\mathcal{T}_{i+1}})^\top, \tag{6}$$

$$s^b_j = \frac{\exp^{\sum_{k=1}^n \mathbf{M}^b_{[j,k]}}}{\sum_{j'=1}^m \exp^{\sum_{k=1}^n \mathbf{M}^b_{[j',k]}}}. \tag{7}$$

The last step produces the final image representation during the backward propagation:

$$\tilde{\mathbf{V}}^b_{\mathcal{T}_i} = s^b \odot \mathbf{V}^b_{\mathcal{T}_i}. \tag{8}$$

Similarly, we use $\tilde{\mathbf{V}}^b_{\mathcal{T}_i} = \mathcal{G}_{\text{MTC}-b}(\mathbf{V}, \tilde{\mathbf{T}}^b_{\mathcal{T}_{i+1}}, \tilde{\mathbf{V}}^b_{\mathcal{T}_{i+1}})$ to denote all computations for the backward MTC gate. When the task corresponds to the third subtask $\mathcal{T}_3$, we use $\tilde{\mathbf{T}}^b_{\mathcal{T}_4} = \mathbf{T}$ and $\tilde{\mathbf{V}}^b_{\mathcal{T}_4} = \mathbf{V}$ as the input.

### 3.5 Joint Multi-Task Multimodal Learning

To jointly learn the three cascaded tasks in a multimodel setting, we propose a Bidirectional Multi-task Cascaded multimodal Fusion framework integrating the task-specific multimodal transformers and the cascading gate respectively introduced in Section 3.3 and 3.4. Specifically, the model takes a pair of text-image $(\mathbf{T}, \mathbf{V})$ as the input to all three task modules. In a forward pass, we first use the forward MTC gate to produce an output image representation for task $\mathcal{T}_1$: $\tilde{\mathbf{V}}^f_{\mathcal{T}_1} = \mathcal{G}_{\text{MTC}-f}(\mathbf{V}, \mathbf{T}, \mathbf{V})$. Then a multimodal transformer block is applied on top of $\tilde{\mathbf{V}}^f_{\mathcal{T}_1}$ and the input text features $\mathbf{T}$ to obtain the output text representation corresponding to $\mathcal{T}_1$: $\tilde{\mathbf{T}}^f_{\mathcal{T}_1} = \mathcal{M}_1(\tilde{\mathbf{V}}^f_{\mathcal{T}_1}, \mathbf{T})$. Subsequently, we obtain outputs for $\mathcal{T}_i$ ($i = 2, 3$) sequentially via $\tilde{\mathbf{V}}^f_{\mathcal{T}_i} = \mathcal{G}_{\text{MTC}-f}(\mathbf{V}, \tilde{\mathbf{T}}^f_{\mathcal{T}_{i-1}}, \tilde{\mathbf{V}}^f_{\mathcal{T}_{i-1}})$ and $\tilde{\mathbf{T}}^f_{\mathcal{T}_i} = \mathcal{M}_i(\tilde{\mathbf{V}}^f_{\mathcal{T}_i}, \mathbf{T})$.

In the backward pass, we first apply the backward MTC gate on the most fine-grained task $\mathcal{T}_3$ to produce an output image representation: $\tilde{\mathbf{V}}^b_{\mathcal{T}_3} = \mathcal{G}_{\text{MTC}-b}(\mathbf{V}, \mathbf{T}, \mathbf{V})$. Then a multimodal transformer block is applied on top of $\tilde{\mathbf{V}}^b_{\mathcal{T}_3}$ and the input text features $\mathbf{T}$ to obtain the output text representation corresponding to $\mathcal{T}_3$: $\tilde{\mathbf{T}}^b_{\mathcal{T}_3} = \mathcal{M}_3(\tilde{\mathbf{V}}^b_{\mathcal{T}_3}, \mathbf{T})$. Subsequently, we obtain outputs for $\mathcal{T}_i$ ($i = 2, 1$) sequentially via $\tilde{\mathbf{V}}^b_{\mathcal{T}_i} = \mathcal{G}_{\text{MTC}-b}(\mathbf{V}, \tilde{\mathbf{T}}^b_{\mathcal{T}_{i+1}}, \tilde{\mathbf{V}}^b_{\mathcal{T}_{i+1}})$ and $\tilde{\mathbf{T}}^b_{\mathcal{T}_i} = \mathcal{M}_i(\tilde{\mathbf{V}}^b_{\mathcal{T}_i}, \mathbf{T})$. The final prediction for each task $\mathcal{T}_i$ is made by concatenating the output text features from both directions:

$$\tilde{\mathbf{y}}_i = \mathcal{F}([\tilde{\mathbf{T}}^f_{\mathcal{T}_i}; \tilde{\mathbf{T}}^b_{\mathcal{T}_i}]). \tag{9}$$

Here $\mathcal{F}$ indicates two linear layers.

During training, the loss of $\mathcal{T}_1$ is:

$$L_{\mathcal{D}_1} = \frac{1}{N_1} \sum_{j=1}^{N_1} L_{\mathcal{T}_1}(X_j, y_j), \tag{10}$$

where each $L_{\mathcal{T}_1}(X_j, y_j)$ computes a cross-entropy loss between the prediction $\tilde{\mathbf{y}}_j$ and the ground-truth label $y_j$ for each instance, and $\mathcal{N}_1$ denotes the total number of training samples in $\mathcal{T}_1$.

Similarly, the losses of $\mathcal{T}_2$ and $\mathcal{T}_3$ are:

$$L_{\mathcal{D}_2} = \frac{1}{N_2} \sum_{j=1}^{N_2} L_{\mathcal{T}_2}(X_j, y_j), \tag{11}$$

$$L_{\mathcal{D}_3} = \frac{1}{N_3} \sum_{j=1}^{N_3} L_{\mathcal{T}_3}(X_j, y_j). \tag{12}$$

We further introduce three hyper-parameters $\lambda$, $\gamma$ and $\sigma$, considering that each task contributes differently to the final outcome. Thus, our objective is to minimize the following aggregated loss:

$$L_{\mathcal{D}} = \lambda L_{\mathcal{D}_1} + \gamma L_{\mathcal{D}_2} + \sigma L_{\mathcal{D}_3}. \tag{13}$$

## 4 Experiments

In this section, we design comprehensive experiments to answer the following four research questions:

- **RQ1:** How does our proposed BiMCF perform on damage analysis compared to other baselines? Why does our BiMCF arrange subtasks from coarse to fine granularity?
- **RQ2:** Can the MTC forward gate and backward gate achieve zoom-in and zoom-out effects respectively? Do they facilitate information complementarity?
- **RQ3:** How does our bidirectional propagation mechanism perform compared to unidirectional propagation?
- **RQ4:** Does our proposed BiMCF demonstrate strong robustness?

We also analyze the sensitivity of our BiMCF to hyper-parameters and the layer number of bidirectional flow in the appendix.

### 4.1 Implementation Details

**Datasets.** Following the prior works on multimodal damage analysis [1, 2, 19, 21], we adopt the standard CrisisMMD benchmark to validate our approach. The samples in this dataset are multimodal social media posts consisting of image-text pairs, which are crawled from Twitter in times of seven natural disasters (*e.g.*, floods, wildfires, hurricanes, earthquakes). Moreover, this benchmark consists of three cascaded subtasks, *i.e.*, informativeness prediction (task1), events classification (task2), severity prediction (task3). The class label of the latter task depends on that of the former task.

**Settings.** We employ pre-trained BERT-base and ViT as our language and image backbones to generate 768-dimensional features. The image input is first resized to 224×224 and then divided into 16×16 patches. During the training process, Adam is adopted as the optimizer with a fixed learning rate of 0.000015. We use a batch size of 32 and train the model for 20 epochs. Moreover, the dropout rate is set as 0.8 and the hyper-parameters are determined by the validation set. All the experiments are run on four A100 GPUs.

**Baselines.** The proposed approach is compared with several existing state-of-the-art baselines, including Cross-Attention [1], Crisis-DIAS [2], interactive multi-task learning network (IMN) [12], Aspect-oriented Method (AoM) [51], Dual Query Prompt Sentiment Analysis (DQPSA) [33], multimodal joint learning (JML) [15], VL-BERT [42], Vision and Language Transformer (ViLT) [16], Qwen-VL [7] and CogVLM [45]. Among them, Cross-Attention filters useless information to achieve effective multimodality fusion. AoM, DQPSA, Crisis-DIAS, IMN and JML are specifically designed for multimodal multi-task learning. VL-BERT and ViLT are typically pretrained vision and language models, which can quickly adapt to downstream tasks by fine-tuning. Two multimodal large language models (MLLMs), Qwen-VL and CogVLM, are also incorporated.

### 4.2 Performance Comparison (RQ1)

We compare our BiMCF with the above mentioned baselines and present the experimental results on the CrisisMMD benchmark

in Table 1. To keep consistent with previous works, we evaluate the performance on each task using the following three metrics: classification accuracy, Macro F1-score and weighted F1-score. Additionally, we conduct comprehensive experiments across all possible task sequences to demonstrate the superiority of our adopted task sequence and display the results in Table 2.

- **Obs1: Our proposed BiMCF achieves state-of-the-art results across all the adopted baselines, demonstrating satisfactory performance improvements compare to both single_task approaches and multi_task approaches (parallel and cascaded).** From Table 1, we can draw the following observations. First, we find that the same model generally performs better in the single_task setting than in the parallel multi_task setting, suggesting that parallel subtasks can negatively interfere with each other. Although existing cascaded multi_task methods take the task interdependencies into account, they still fail to outperform our approach. This can be attributed to that our BiMCF fully explores the underlying relationships between the cascaded tasks and the bidirectional flows effectively suppress the propagation of negative messages, thus enabling a further augmentation of the multimodal fusion capability.
- **Obs2: Our proposed BiMCF exhibits superior capability in disaster analysis compared to MLLMs** We also make a performance comparison between our approach and two currently popular MLLMs, Qwen-VL and CogVLM. Specifically, we utilize text prompts to enable Qwen-VL and CogVLM to perform damage predictions. From Table 1, we can observe that without fine-tuning, there remains a significant performance gap between the two MLLMs and ours. Then we apply LoRA [14] to fine-tune Qwen-VL and CogVLM, yielding performance improvements of 4.3% to 10.8% in multi_task across Acc metric. However, they still don't surpass the performance of our model.
- **Obs3: Arranging the three subtasks from coarse to fine granularity facilitates a gradual understanding of complex information and outperforms other possible task sequences.** From Table 2, it is evident that our sequential approach achieves the best performance, with an average improvement of 2.1% to 4.9% across all tasks. In the CrisisMMD dataset, only informative tweets are used to determine the type of disaster, after which the severity is analyzed based on that classification. Therefore, our adopted sequence seamlessly aligns with the intrinsic interdependencies among the three subtasks and can prove to be preferable to other task sequences.

### 4.3 Visualization (RQ2)

In this part, we present the visualization of the attended image features in different subtasks and pass mechanisms in Figure 3.

- **Obs4: The MTC forward gate achieves the zoom-in effect to keep salient features, while the MTC backward gate achieves the zoom-out effect to obtain rich global information.** As shown in Figure 3, in the forward flow, we can observe from top to bottom that the attended image regions are progressively narrowed down to specific areas. This is because later tasks are more fine-grained and focus more on specific details. Similarly, in the backward flow, we can observe from bottom to up that the attended areas expand around some anchor regions.

Table 1: Performance comparison with baselines on CrisisMMD. 'Single_task' denotes treating each subtask independently. 'Parallel multi_task' and 'Cascaded multi_task' address all three subtasks simultaneously. However, the former doesn't consider the inherent dependencies between subtasks, while the latter does. The best results are in bold, and the second-best results are underlined. Metrics include Acc: classification accuracy, M-F1: Macro F1-score and W-F1: weighted F1-score.

| Category | Method | Task 1 | | | Task 2 | | | Task 3 | | |
|---|---|---|---|---|---|---|---|---|---|---|
| | | Acc(%) | M-F1(%) | W-F1(%) | Acc(%) | M-F1(%) | W-F1(%) | Acc(%) | M-F1(%) | W-F1(%) |
| Single_task | Cross-Attention [1] | 83.7 | 83.1 | 83.6 | 80.7 | 74.6 | 81.1 | 88.9 | 51.6 | 88.5 |
| | VLBert [42] | 84.1 | 83.9 | 84.0 | 81.7 | 75.2 | 81.9 | 90.9 | 55.7 | 90.3 |
| | ViLT [16] | 83.6 | 83.3 | 83.4 | 82.1 | 74.7 | 81.9 | 90.2 | 55.3 | 89.7 |
| | Qwen-VL [7] | 75.2 | 74.6 | 75.1 | 71.7 | 69.2 | 71.5 | 86.8 | 50.8 | 86.6 |
| | Qwen-VL+LoRA [14] | 76.3 | 75.9 | 76.2 | 73.6 | 71.3 | 73.8 | 88.1 | 51.6 | 87.9 |
| | CogVLM [45] | 77.3 | 76.7 | 77.1 | 70.6 | 67.4 | 69.8 | 87.4 | 54.6 | 86.1 |
| | CogVLM+LoRA | 79.1 | 79.0 | 78.9 | 72.4 | 68.2 | 71.3 | 89.1 | 55.9 | 88.6 |
| Multi_task | Parallel Multi_task | | | | | | | | | |
| | Cross-Attention | 80.7 | 80.1 | 80.4 | 79.7 | 72.3 | 79.1 | 86.9 | 50.8 | 85.9 |
| | VLBert | 81.9 | 81.3 | 81.7 | 80.7 | 72.7 | 80.2 | 89.2 | 54.1 | 89.0 |
| | ViLT | 81.2 | 80.9 | 81.0 | 80.1 | 72.6 | 79.8 | 88.7 | 54.6 | 88.1 |
| | Qwen-VL | 72.6 | 72.3 | 72.5 | 70.1 | 68.3 | 69.7 | 84.2 | 51.9 | 83.6 |
| | Qwen-VL+LoRA | 80.3 | 80.4 | 80.3 | 76.2 | 70.7 | 76.5 | 88.5 | 53.8 | 88.2 |
| | CogVLM | 75.3 | 75.1 | 75.3 | 68.1 | 75.4 | 67.7 | 85.5 | 53.6 | 85.1 |
| | CogVLM+LoRA | 81.9 | 81.7 | 81.8 | 78.9 | 76.1 | 78.5 | 89.6 | 54.1 | 89.4 |
| | IMN [12] | 81.8 | 81.5 | 81.8 | 80.7 | 77.1 | 80.4 | 90.2 | 53.9 | 89.7 |
| | Cascaded Multi_task | | | | | | | | | |
| | DQPSA [33] | 82.3 | 82.1 | 82.2 | 76.8 | 74.6 | 76.7 | 90.7 | 56.1 | 91.1 |
| | AoM [51] | 83.1 | 82.8 | 82.9 | 80.6 | 76.1 | 77.1 | 91.1 | 56.3 | 90.9 |
| | Crisis-DIAS [2] | 84.0 | 83.7 | 83.9 | 77.5 | 75.7 | 77.9 | 90.4 | 55.6 | 90.9 |
| | JML [15] | 82.5 | 81.9 | 82.2 | 81.4 | 78.2 | 80.9 | 91.2 | 51.9 | 89.9 |
| | BiMCF (ours) | 87.1 | 87.1 | 87.1 | 84.2 | 78.2 | 84.3 | 93.4 | 57.0 | 93.7 |

Table 2: Task sequence analysis across Acc (%) metric. The task sequence '1->2->3' and '3->2->1' are equivalent due to the bidirectional propagation of our BiMCF.

| Task Sequence | | Task1 | Task2 | Task3 |
|---|---|---|---|---|
| 1->2->3
3->2->1 | (ours) | 87.1±0.3 | 84.2±0.4 | 93.4±0.3 |
| 1->3->2
2->3->1 | | 85.9±0.4 | 80.2±0.4 | 90.7±0.2 |
| 2->1->3
3->1->2 | | 84.9±0.3 | 81.8±0.5 | 91.3±0.3 |

This is because preceding tasks are more coarse-grained and require abundant global information.

- **Obs5: The combined MTC forward & backward gate can comprehensively locate task-specific features and avoid locating helpless features.** Looking from left to right in Figure 3, we can observe that the forward and backward gates have their respective advantages and are able to attend to the image features that the other overlooks. However, they fail to locate all task-related features and attend to some irrelevant features. Fortunately, the combination of them helps alleviate these issues.

## 4.4 Ablation Study (RQ3)

In order to validate the superiority of our bidirectional cascade, we conduct comprehensive ablation study on the CrisisMMD benchmark and display the results in Table 3.

- **Obs6: Our bidirectional propagation mechanism alleviates errors accumulated in one direction and allows for mutual promotion in forward and backward flow, thereby attaining better results.** As shown in the first row in Table 3, the remarkable results demonstrate the effectiveness of our bidirectional propagation mechanism. In the next two rows, we remove the forward flow and backward flow from our model respectively. It clearly shows that there has been a certain degradation of the performance. We further carry out experiments without incorporating forward and backward flows and see that the results get worse as expected. From the above observations, we can conclude that while forward flow and backward flow can independently enhance the model's performance, their combination allows them to exchange information more effectively, leading to further performance improvement.

## 4.5 Robustness Analysis (RQ4)

Table 4 presents the performance on the three tasks when employing different text and image backbones.

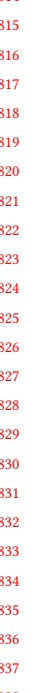

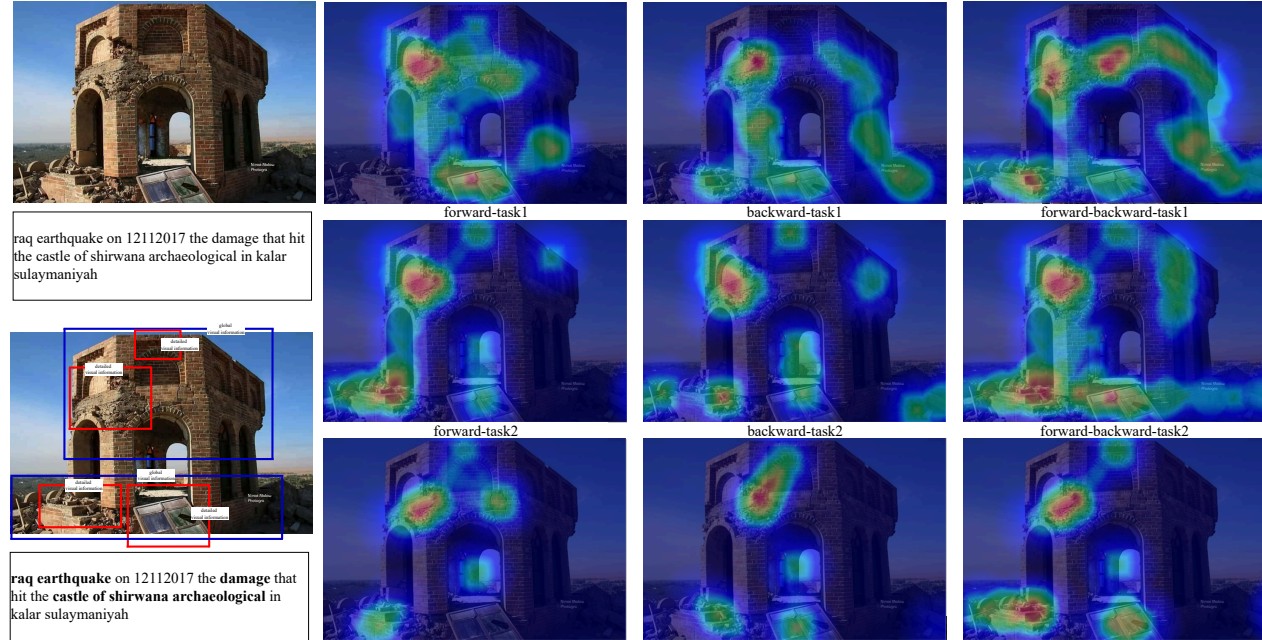

**Figure 3: Visualization of the attended image information in the three substasks, considering different pass mechanisms: forward pass only, backward pass only and forward & backward pass. Given a text-image pair, the vertical arrangement symbolizes the attended image information by different tasks after passing through the cascaded gates. The horizontal arrangement highlights the comparison of attended information among the three propagation mechanisms.**

**Table 3: Ablation study. The notation '-forward flow' means removing forward flow from our BiMCF model, and so on.**

| Model design | Task 1 | | | Task 2 | | | Task 3 | | |
|---|---|---|---|---|---|---|---|---|---|
| | Acc(%) | M-F1(%) | W-F1(%) | Acc(%) | M-F1(%) | W-F1(%) | Acc(%) | M-F1(%) | W-F1(%) |
| BiMCF | **87.1** | **87.1** | **87.1** | **84.2** | **78.2** | **84.3** | **93.4** | **57.0** | **93.7** |
| -forward flow | 85.1 | 84.7 | 85.3 | 81.7 | 75.1 | 81.9 | 90.7 | 55.0 | 90.6 |
| -backward flow | 84.9 | 84.8 | 84.8 | 82.3 | 75.5 | 82.1 | 91.3 | 55.6 | 91.4 |
| -forward & backward flow | 81.2 | 81.3 | 81.2 | 78.4 | 71.9 | 77.9 | 89.7 | 50.8 | 89.1 |

**Table 4: The performance of our model on the combination of different text and image encoders across Acc (%) metric.**

| Text Encoder | Image Encoder | Task1 | Task2 | Task3 |
|---|---|---|---|---|
| BERT-base | ViT | 87.1±0.3 | 84.2±0.4 | 93.4±0.3 |
| XLNet [47] | ViT | 88.2±0.2 | 84.5±0.4 | 93.7±0.3 |
| RoBERTa [22] | ViT | 87.6±0.1 | 84.2±0.3 | 93.1±0.2 |
| BERT | CLIP [37] | 86.8±0.3 | 84.4±0.3 | 93.0±0.4 |
| BERT | Swin-T [23] | 87.3±0.5 | 84.7±0.3 | 93.9±0.3 |

- **Obs7: Our proposed BiMCF demonstrates superior robustness.** From Table 4, we can clearly observe that our approach can consistently achieve excellent performance with the combinations of different text and image backbones. When we utilize more powerful feature extractors, our model can obtain better

results than those reported in Table 1. This observation indicates that our proposed BiMCF can effectively adapts to various backbones without compromising its accuracy.

## 5 Conclusion

This work proposes the bidirectional multi-task cascaded multimodal fusion approach towards joint multimodal damage analysis. Based on this, we introduce a bidirectional multi-task cascaded framework, which ranks the three related subtasks based on the increasing granularity. Through incorporating forward and backward flow, this framework propagates information in dual directions and exchanges information more effectively between each subtask. Considering that different tasks attend to different image information, we further present a unified multimodal fusion framework to effectively integrate information from both textual and visual modalities. We evaluate this approach on the three subtasks of damage analysis, and obtain superior results across all the baselines.

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

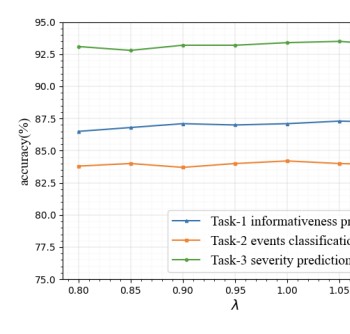 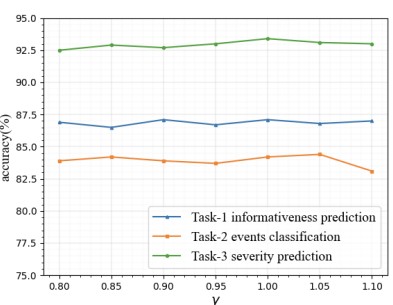 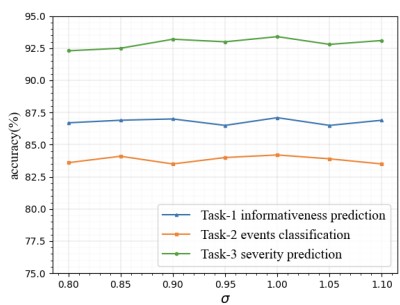

**Figure 4: Sensitivity analysis for the hyper-parameters($\lambda$, $\gamma$ and $\sigma$) in terms of classification accuracy. We conduct the experiments by changing the value of one hyper-parameter within a certain range, while setting the values of other hyper-parameters to a constant 1. All the three subtasks are included into the analysis experiment.**

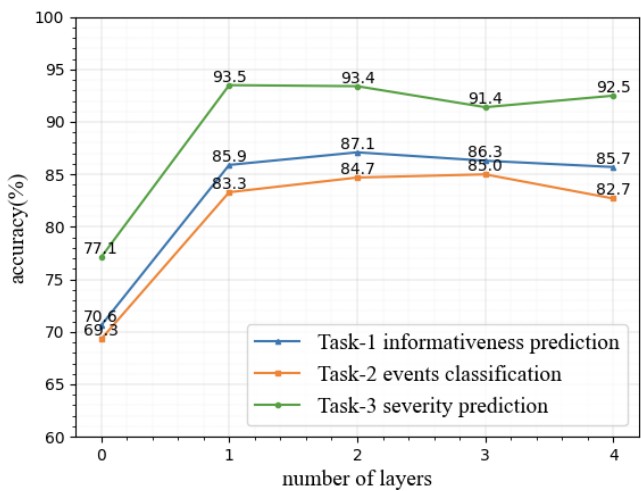

**Figure 5: Analysis on the layer number of bidirectional flow in terms of classification accuracy.**

## A APPENDIX

## A.1 Supplementary Experimental Results

*A.1.1 Hyper-parameter Sensitivity.* In this section, we analyze the sensitivity of our BiMCF to variations about hyper-parameters and display the result in Figure 4.

- **Obs8: The performance of our BiMCF is not sensitive to the variations of hyper-parameters.** We examine the impact on the identification of the three subtasks when changing the trade-off hyper-parameter $\lambda$ for $L_{\mathcal{D}_1}$, $\gamma$ for $L_{\mathcal{D}_2}$ and $\sigma$ for $L_{\mathcal{D}_3}$ respectively. Taking the experiment on $\alpha$ in Figure 4 as an example, we change the value of $\lambda$ within a certain range and keep the value of other hyper-parameters as the constant 1. We can easily find that the three curves, representing the classification accuracy of the three subtasks, exhibits minimal fluctuations and keep smooth, which means that our method is not sensitive to $\lambda$. Similarly, we can also find that our BiMCF is not sensitive to $\gamma$ and $\sigma$ as well. Therefore, the above analysis fully demonstrates the robustness of our proposed approach.

*A.1.2 Layer Analysis.* In this section, we investigate how the number of bidirectional flow influence the performance of our model and the corresponding results on the three subtasks are shown in Figure 5.

- **Obs9: When the number of the adopted bidirectional flow is set to 2, the model achieves optimal overall performance.** From Figure 5, we can observe that the performance of our approach is definitely improved when the bidirectional flow is inserted to the model, indicating that richer features are extracted and better modality fusion is achieved. With the increase of the layer number, the performance of the model gradually improves at first, and then begins to decrease after reaching the best. Therefore, in our experiment, we adopt two bidirectional flows, which enables our proposed approach to achieve the optimal performance overall.

