# OpenReview forum: "Damage Analysis via Bidirectional Multi-Task Cascaded Multimodal Fusion"
_ACM.org/TheWebConf/2025/Conference — WWW 2025 Oral_

### Official Review · Reviewer_MWWN · 2024-11-20

**Novelty:** 5
**Technical Quality:** 5

**Review:**

*Pros*
- The reseach problem of damage analysis is of practical value.
- This work notes that existing methods address each damage-related subtask individually and propose an approach to address this issue.
- The experimental results show that the proposed method has a promising performance.

*Cons*
- The source code could be provided to facilitate the reproducibility of this work. The site https://anonymous.4open.science/ could be helpful.
- This work only considers one dataset. It could be better to verify the effectiveness of the proposed method in another dataset from a different domain.
- In my view, Figure 1 is not very typical of the issue. It seems that both the global information and the detailed information are important to both subtasks. Moreover, it only reflects the image modal, ignoring the other modalities.

**Questions:**

The same as the cons.

**Reviewer Confidence:**

2: The reviewer is willing to defend the evaluation, but it is likely that the reviewer did not understand parts of the paper

**Scope:**

3: The work is somewhat relevant to the Web and to the track, and is of narrow interest to a sub-community

---

### Official Review · Reviewer_LgdB · 2024-11-26

**Novelty:** 6
**Technical Quality:** 5

**Review:**

The authors propose the Multi-Task Cascaded Multimodal Fusion (BiMCF) approach to determining the physical damage contained within images. They identify three related tasks that form the approach: informativeness, event classification, and severity. The authors clearly explore the existing research and emerging trends within damage analysis. They insightfully describe the application and novelty of their approach. Finally, the authors convincingly demonstrate the efficacy of their approach through a comprehensive analysis of competing state-of-the-art approaches.

Pros:
- Well referenced research of existing research and emerging trends.
- Good explanation of damage analysis and the existing works.
- Very clear description of the informativeness, event and severity classifications.
- The use of bi-directional cascading is very interesting and its necessity is well justified.
- The methodology is explained clearly and concisely, despite containing dense formal definitions.
- The results are impressive and demonstrate the effectiveness of the approach.
- Comprehensive comparison to the state-of-the-art alternative approaches.
- Figures and Tables provide insight into the research.

Cons:
- Some grammatical and spelling errors.
- The introduction & abstract lack a clear motivation behind the work.
- The Formal definitions are quite complex and hard to follow for a non-expert in multi-modal models.
- The paper's relevance to social media & social networks is noted; however, the research is not intrinsic to social networks. The paper's relevance seems to align better with the broader category of image analysis.

Importantly, feedback cannot be provided regarding the accuracy of formal definitions as the reviewer is not an expert on transformer models or their related components.

**Questions:**

Abstract: it is unclear why damage analysis is useful.

Introduction - Line 152: Consider including a short description of  "BERT"

I did want to mention that the Methodology section is excellent at explaining the approach and the necessity for every step in the framework. As a non-expert in multi-modal models, this was very easy to follow. Except for the dense formalizations, however, I do not believe there is a way to circumvent that complexity.

Finally, there are a couple of minor grammatical issues scattered throughout the paper, I would encourage the authors to perhaps analyze the paper more rigorously; however, these are some of the errors on the first page, which I have noted down.
Introduction - Line 96: "happened all around the world" should be "happening all around the world"
Introduction - Line 105 & 196: "as standard benchmark" should be "as [a] standard benchmark" or "as [the] standard benchmark". This aligns with how CrisisMMD is described on Line 115 & 605
Introduction - Line 110: "where an posted blog" should be "where a posted blog"
Conclusion - Line 916: Make sure the references to the BiMCF approach are uniform in formatting.

Also, make sure to update the conference information.

**Reviewer Confidence:**

2: The reviewer is willing to defend the evaluation, but it is likely that the reviewer did not understand parts of the paper

**Scope:**

3: The work is somewhat relevant to the Web and to the track, and is of narrow interest to a sub-community

---

### Official Review · Reviewer_43rs · 2024-12-01

**Novelty:** 5
**Technical Quality:** 5

**Review:**

This research aims to address damage analysis, which is important for maintaining information security on the Internet. In addition, the authors propose a Bidirectional Multi-Task Cascaded Multimodal Fusion (BiMCF) approach to implement comprehensive multimodal damage analysis from the multi-task learning perspective, with the purpose of exploiting the potential relationships between tasks. Specifically, the authors propose the BiMCF approach by introducing the bidirectional cascaded framework to implement multimodal multi-task learning, which encodes the independencies among subtasks into the model structure and enhances the multimodal fusion process. Finally, comprehensive experiments on the CrisisMMD benchmark are conducted to show the effectiveness of our approach.
But there are still some concerns.
1, I think figure 2 could be better. The authors are advised to pay attention to the use of colors, and one color should try to indicate one module. The red module on the left of Fig. 2 indicates a multi-task forward propagation gate, but the red module in the right figure indicates a multi-head attention mechanism, which I think is not rigorous. In addition, there is a straight line misalignment in Figure II.
2, it is suggested that the authors add a detailed description of the dataset. It is not clear to me what the labels are for each of the three subtasks in the dataset.
3, the authors only experimented under one dataset. It is recommended that the authors experiment under multiple datasets.
4, It is recommended that the authors add the necessary hyperparameters for the experiment. It is not clear to me whether the weight coefficients of different subtasks have any effect on the model effect. The authors also did not label the values of different subtask weights in the implementation details.

**Questions:**

1, What are the labels of each subtask in the dataset?
2, Could the authors analyze why the effect of most of the large multimodal language models performs poorly on this task.
3, Could the authors analyze if the setting of different subtask weights has any effect on the model effectiveness?

**Reviewer Confidence:**

3: The reviewer is confident but not certain that the evaluation is correct

**Scope:**

4: The work is relevant to the Web and to the track, and is of broad interest to the community

---

### Official Review · Reviewer_NAb9 · 2024-12-02

**Novelty:** 5
**Technical Quality:** 5

**Review:**

The paper presents a novel approach named Bidirectional Multi-task Cascaded Multimodal Fusion (BiMCF) for damage analysis from multimodal social media data. The methodology integrates visual and textual information to address subtasks arranged in a cascaded structure., so that the limitations of previous works that tackle the subtasks individually are addressed. Specifically, the method applied bidirectional propagation mechanism to enhance multimodal fusion. In this process, task-specific multimodal transformers to account for subtask-specific needs. Finally, to mitigate error accumulation a unique zoom-in (forward) and zoom-out (backward) information flow. Experimental results show powerful performance of the proposed method.

Strengths:
1. The bidirectional multimodal fusion with cascading subtasks effectively captures task-specific interdependencies and enables information sharing across subtasks.

2. The work rigorously evaluates the approach on the CrisisMMD benchmark and outperforms existing state-of-the-art methods across multiple evaluation metrics.

3. Clear visualizations of the attention mechanism highlight how forward and backward propagation contribute to feature selection and fusion.

4. The adaptability of BiMCF to various backbone models (e.g., BERT, ViT, RoBERTa) demonstrates its flexibility and robustness.

**Questions:**

1. The evaluation relies heavily on the CrisisMMD dataset.Is there any other results tested on additional datasets or real-world scenarios?

**Reviewer Confidence:**

3: The reviewer is confident but not certain that the evaluation is correct

**Scope:**

3: The work is somewhat relevant to the Web and to the track, and is of narrow interest to a sub-community